

# MicroRNA-705 regulates the differentiation of mouse mandible bone marrow mesenchymal stem cells

Xiao Hong Yang[1,*], Kun Yang[2,*], Yu Lin An[3], Li Bo Wang[1,4], Guo Luo[4] and Xiao Hua Hu[5]

[1] Department of Prosthetics, the Affiliated Stomatology Hospital of Zunyi Medical University, Zunyi Medical University, Zunyi, Guizhou, China
[2] Department of Periodontology, the Affiliated Stomatology Hospital of Zunyi Medical University, Zunyi Medical university, Zunyi, Guizhou, China
[3] Department of Stomatology, Jinling Hospital, Medical School of Nanjing University, Nanjing, Jiangsu, China
[4] Department of Stomatology, Zunyi Medical University, Zunyi, Guizhou, China
[5] Department of Oral and Maxillofacial Surgery, the Affiliated Stomatology Hospital of Zunyi Medical University, Zunyi, Guizhou, China
* These authors contributed equally to this work.

Corresponding author
Xiao Hua Hu, hxh2132000@163.com

## ABSTRACT

The craniofacial skeleton is the foundation of most stomatological treatments, including prosthodontics and maxillofacial surgery. Although histologically similar to the appendicular skeleton, the craniofacial skeleton manifests many unique properties in response to external stimuli and signals. However, the mandibular or maxillary bone marrow mesenchyme, which is the intrinsic foundation of the functions of craniofacial skeleton, has not been well studied, and its homeostasis mechanism remains elusive. Osteoporosis is a systemic disease that affects all skeletons and is characterized by bone mass loss. Osteoporotic bone marrow mesenchymal stem cells (BMMSCs) exhibit disturbed homeostasis and distorted lineage commitment. Many reports have shown that microRNAs (miRNAs) play important roles in regulating MSCs homeostasis. Here, to obtain a better understanding of mandibular bone marrow MSCs homeostasis, we isolated and cultured mandible marrow MSCs from mouse mandibles. Using miR-705 mimics and an inhibitor, we demonstrated that miR-705 played a vital role in shifting the mandibular MSCs lineage commitment *in vitro*. Utilizing an osteoporosis mouse model, we demonstrated that MSCs from ovariectomized (OVX) mouse mandibular bone marrow exhibited impaired osteogenic and excessive adipogenic differentiation. miR-705 was found overexpressed in OVX mandibular MSCs. The knock down of miR-705 *in vitro* partially attenuated the differentiation disorder of the OVX mandibular MSCs by upregulating the expression of osteogenic marker genes but suppressing adipogenic genes. Taken together, our findings provide a better understanding of the homeostasis mechanism of mandibular BMMSCs and a novel potential therapeutic target for treating mandibular osteoporosis.

## INTRODUCTION

Bone marrow homeostasis is vital for the function of bone marrow mesenchymal stem cells (BMMSCs) and thus the function of bones. Disturbance of homeostasis may result in a shift in cell lineage commitment of BMMSCs and cause systemic diseases, such as osteoporosis and hyperparathyroidism (*Simonds et al., 2002*). BMMSCs from osteoporotic bone marrow exhibit reduced osteogenic and enhanced adipogenic potential, resulting in constant bone mass loss and a decline in mechanical strength in all bones, including the craniofacial skeleton (*Chen et al., 2016*; *Jonasson & Rythen, 2016*; *Mavropoulos, Rizzoli & Ammann, 2007*). Despite many reports on this topic, the mechanisms behind the shift remain elusive.

Although the craniofacial, axial and appendicular skeletons share the same histological morphology, the craniofacial skeleton has a different homeostasis mechanism than those of the axial and appendicular skeletons. Unlike the axial and appendicular skeletons, the craniofacial skeleton is generated by neural crest cells from the neuroectoderm germ layer (*Li, Parada & Chai, 2017*; *Minoux & Rijli, 2010*) and undergoes intra-membranous ossification instead of endochondral ossification (*Minarikova et al., 2015*). Differences can also be observed in the responses of these skeletons to external stimuli. Systemic diseases, such as cherubism (*Brix, Peters & Lebeau, 2009*) and bisphosphonate-related osteonecrosis of the jaws (*Dimitrakopoulos, Magopoulos & Karakasis, 2006*; *Migliorati et al., 2005*), affect only alveolar bones. Previous studies also showed that alveolar bones lost less trabecular bone mass at a lower rate in a rodent osteoporosis model, which was quite different from the observations in long bones (*Mavropoulos, Rizzoli & Ammann, 2007*). This evidence led us to hypothesize that MSCs in the mandibular or maxillary bone marrow might possess some unique characters in cell lineage commitment and homeostasis sustaining. In this study, we constructed a mouse osteoporosis model to study the differentiation properties and regulatory mechanism of mandibular MSCs. We found that mandibular MSCs in the model mouse exhibited lower osteogenic potential but higher adipogenic potential through *ex vivo* induction and staining. RNA expression of marker genes provided similar conclusions. Subsequently, we focused on microRNAs (miRNAs) to uncover the regulatory mechanism of mandibular MSCs.

miRNAs function by base pairing to complementary sites in the 3′ untranslated region (3′UTR) of mRNAs (*Ab Mutalib et al., 2016*; *Bartel, 2009*). The connections between miRNAs and the cell lineage commitment of MSCs have been demonstrated by several studies using BMMSCs from femurs (*Chen et al., 2016*). In our previous study, we identified miR-705 in mouse femur BMMSCs as a negative regulator of osteoblast differentiation (*Liao et al., 2013*). Overexpression of miR-705 caused extra bone mass loss in a mouse model, whereas its knockdown attenuated bone loss in an ovariectomized (OVX) mouse model.

Here, based upon our previous discovery, we confirmed that miR-705 exerted a negative regulatory function in mouse mandibular MSCs. In the gain and loss assay, we showed that miR-705 played a key role in regulating lineage commitment in MSCs from OVX mouse mandibles. Knocking down miR-705 in OVX mandibular MSCs resulted in elevated osteoblast differentiation but reduced adipocyte differentiation *in vitro*. Thus, our study

suggested a novel regulatory mechanism in mandibular MSCs that might promote the discovery of new therapeutic targets for osteoporotic alveolar bones.

## MATERIALS & METHODS

### Animals

Sixty 6-week-old female C57BL/6J mice were randomly divided into two groups. Mice in the Sham group underwent sham surgery. Mice in the OVX group received a bilateral OVX under general anaesthesia. After surgery, all mice were housed under pathogen-free conditions (24 °C, 12-hour light/12-hour dark cycles and 50–55% humidity) for 3 months. All animal procedures were performed according to the guidelines of the Animal Care Committee of Zunyi Medical University, Zunyi, Guizhou, China.

### Cell culture

Three months after the OVX and sham surgeries, MSCs were isolated from the mouse mandibles and cultured. Briefly, the mandibles were removed aseptically after euthanasia and dissected free of soft tissues. The periodontal tissue with the whole teeth was removed. Part of the alveolar bone surrounding the tooth socket was also removed to ensure clearance of dental pulp and odontoblast contamination. Then, the mandibles were smashed with scissors and flushed with $\alpha$-MEM (Invitrogen, Carlsbad, CA, USA, supplemented with 10% FBS and 1% penicillin and streptomycin) to free the marrow cavities. The cell suspension was filtered, seeded into 5-cm culture dishes and grown in $\alpha$-MEM in a humidified atmosphere with 5% $CO_2$ at 37 °C. The medium was changed every two days, and adherent cells at confluence were passaged with 0.25% trypsin/1 mM EDTA. Cells from passages 2–3 were used in the experiments.

### Alizarin red staining

Mandibular MSCs at the 2nd passage were digested and seeded in osteogenesis-inducing medium (100 mg/ml ascorbic acid, 2 mM $\beta$-glycerophosphate and 10 nM dexamethasone) for two weeks to induce osteoblast differentiation. Alizarin red staining was performed as described previously (*Yang et al., 2013*). Quantification analysis was performed by extracting mineralized nodules with cetylpyridinium chloride and measuring the absorbance with a spectrophotometer at 570 nm.

### Oil red O staining

Mandibular MSCs at the 2nd passage were digested and seeded in adipogenesis-inducing medium (0.5 mM isobutyl methylxanthine, 0.5 mM dexamethasone and 60 mM indomethacin; Sigma-Aldrich, St. Louis, MO, USA) for one week for adipogenic induction. Oil red O staining was performed to detect lipid droplets as described previously (*Yang et al., 2013*). Quantification analysis was performed by extracting lipid droplets with isopropanol and measuring the absorbance with a spectrophotometer at 520 nm.

### Real-time *q*RT-PCR analysis of mRNAs and miRNAs

Total RNA was extracted using the TRIzol reagent (Invitrogen Life Technology, Carlsbad, CA, USA). Single-strand cDNA was synthesized from 2 mg of total RNA with the

PrimeScript RT reagent kit (TaKaRa Bio, Dalian, China). For the *q*RT-PCR analysis, the miRNAs were reverse-transcribed with RT primers (RiboBio, Guangzhou, China). *β*-actin and U6 were used as loading controls for quantitation of the mRNAs and miRNAs respectively. The Bulge-Loop miRNA qRT-PCR Primer Set (RiboBio) was used for qRT-PCR of miR-705 (product ID: miRQ0003495-1-2) and U6 (Product ID: MQP-0202). All real-time PCR analyses were performed and analysed with the SYBR Premix Ex Taq II kit (TaKaRa) and the ABI Prism 7500 HT sequence detection system (Applied Biosystems, Foster City, CA, USA). The primer sequences are shown in Table S1.

## Western blot

The western blot analyses were performed as previously described (*Liao et al., 2013*). The proteins were loaded on 10% sodium dodecyl sulphate polyacrylamide gels, transferred to polyvinylidene fluoride membranes (Millipore, Billerica, MA, USA) and blocked with 5% non-fat milk powder in PBST (containing 0.1% Tween). The membranes were probed overnight with the following primary antibodies: *β*-actin (Cell Signalling, Beverly, MA, USA), homeodomain-containing factor A10 (HOXA10) (Santa Cruz, Dallas, TX, USA) and Forkhead box O1 (FoxO1) (Cell Signalling). The membranes were incubated with peroxidase-conjugated secondary antibody (Boster, Wuhan, China). The blots were visualized through an enhanced chemiluminescence kit (Amersham Biosciences, Piscataway, NJ, USA) according to the recommended instructions.

## Transfection of the miRNA mimics and inhibitor

The miR-705 mimics, inhibitor and negative control were purchased from RiboBio. The final concentration used for transfection was adjusted to 50 nM. The siPORTNeoFX transfection reagent (Ambion, Austin, TX, USA) was used according to the manufacturer's instruction. Briefly, diluted miRNA mimics/inhibitor and diluted siPORTNeoFX transfection reagent were mixed and incubated for 10 min at room temperature. Cultured cells at 80% confluence were digested, and the suspensions were overlaid onto the transfection complexes and incubated at 37 °C for 48 h. Control group cells received the transfection reagent alone, and the other operations were the same.

## Micro-CT analysis

The mice were scanned with the Inveon micro-CT system (Siemens AG, Germany). The scans were performed using an 80-kV and 500-mA micro-focus X-ray source. The cross-sectional volumetric bone mineral density (BMD) was measured at the angle of the mandible (the rectangle area near the joint area of the body and the ramus of the mandible posterior to the molars). Bone morphometric parameters and the bone volume relative to the tissue volume (BV/TV) were also assessed. Micro-CT scanning of femurs was performed as described previously (*Yang et al., 2013*).

## Statistical analysis

The data are presented as the means ± SDs. Comparisons were made using one-way ANOVA followed by Tukey-Kramer test as a post-hoc. All experiments were repeated at least three times, and representative experiments were shown. Differences were considered significant at $P < 0.05$.

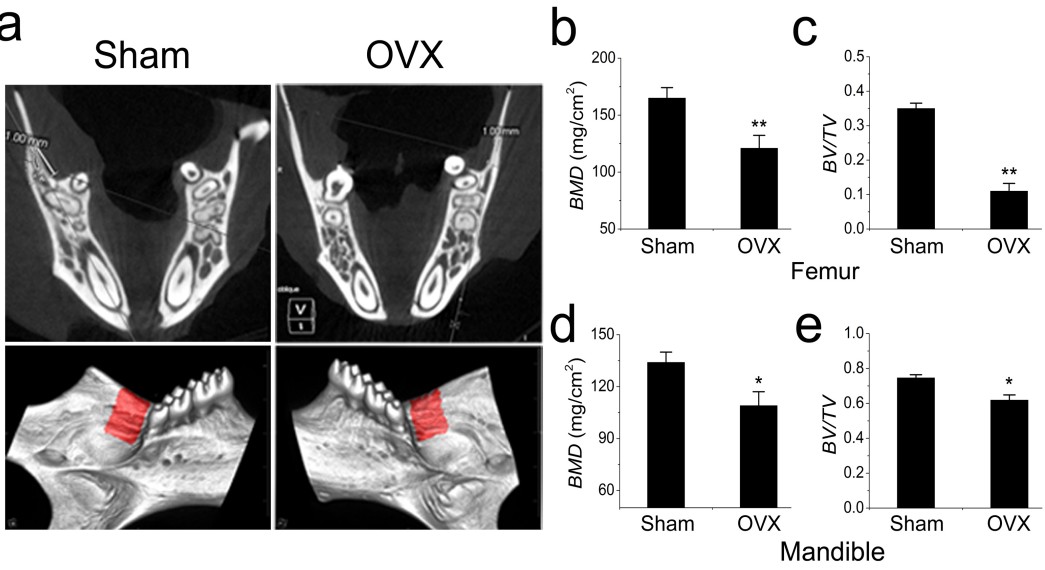

**Figure 1  Micro-CT and quantification of mouse mandible and femur.** (A) Micro-CT analysis of mandibles from the Sham and OVX mice. The red area is the target area for quantification. (B) BMD and (C) BV/TV comparisons of femurs from the Sham and OVX mice. (D) BMD and (E) BV/TV comparisons of mandibles from the Sham and OVX mice. Values are described as the mean ± SD from three independent experiments. * $P < 0.05$, ** $P < 0.01$; Sham, sham surgery; OVX, ovariectomy; BMD, bone mineral density; BV/TV, bone volume/total volume.

# RESULTS

## Cell lineage commitment of mandibular MSCs shifts to adipocytes during osteoporosis

The OVX models were constructed and confirmed as previously described (*Liao et al., 2013*). As expected, all mice manifested osteoporosis 3 months after OVX. In addition to the femur results, we also focused on the mandible (Fig. 1A). The BMD and BV/TV were significantly reduced in the femurs of OVX mouse (Figs. 1B, 1C). However, the BMD and BV/TV comparisons were barely significant between the OVX and Sham mandibles, which were quite different from the results obtained in the long bones (Figs. 1D, 1E).

MSCs were isolated from both the OVX and Sham mouse mandibles and cultured *in vitro* for further investigation. The flow cytometry analysis showed that these cells were all positive for CD29, CD105, CD106 and SCA-1 but were negative for CD34 and CD45. No significant difference was observed in the proliferation comparison.

We then compared the differentiation potentials of the OVX and Sham mandibular MSCs. The OVX mandibular MSCs formed much fewer mineralized nodules but more adipocytes than the Sham MSCs during induction, suggesting that weaker osteogenic but stronger adipogenic differentiation occurred in the OVX mandibular MSCs (Figs. 2A, 2B, 2E and 2F). This result was confirmed by *q*RT-PCR. Runt-related transcription factor 2 (RUNX2) and alkaline phosphatase (ALP) were analysed as two osteogenic marker genes. The OVX mandibular MSCs expressed significantly less RUNX2 and ALP than the Sham MSCs after 2 weeks of osteogenic induction (Figs. 2C, 2D). However, after 1 week

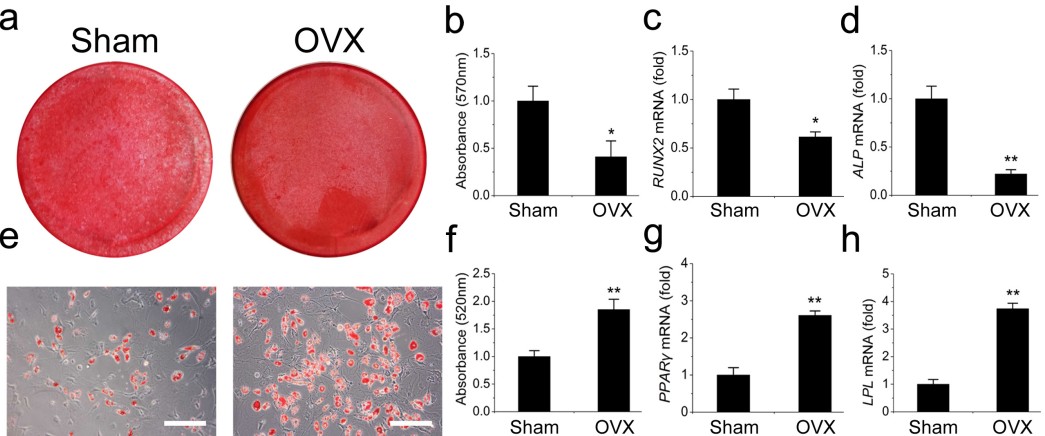

**Figure 2** **Shift of cell lineage commitment in MSCs from osteoporosis mandible bone marrow.** Osteoblast differentiation was impaired in MSCs from osteoporosis mandible bone marrow. MSCs from OVX and Sham mandible at P3 were induced with osteogenic medium for 14 days. Alizarin red staining was performed (A) and quantified (B) via extraction with cetylpyridinium chloride. $q$RT-PCR was used to determine the expression of RUNX2 (C) and ALP (D) in P3 MSCs. Adipocyte differentiation was enhanced in MSCs from osteoporosis mandible. MSCs from OVX and Sham mandible at P3 were induced with adipogenic medium for 7 days. Oil red O staining was performed (E) and quantified (F) via extraction with isopropanol. $q$RT-PCR was used to determine the expression of PPAR-$\gamma$ (G) and LPL (H) in P3 MSCs . The scale bars in micrographs represent 50 $\mu$m. Values are described as mean ± SD from three independent experiments. *$P < 0.05$, **$P < 0.01$. P3, third passage.

of adipogenic induction, the expression of peroxisome proliferator-activated receptor-$\gamma$ (PPAR-$\gamma$) and lipoprotein lipase (LPL), which were adipogenic marker genes, were significantly increased compared with the levels in the Sham MSCs (Figs. 2G, 2H).

## Osteoporotic mandibular MSCs overexpress miR-705

In a previous study, we detected the overexpression of miR-705 in osteoporotic bone marrow using MSCs from OVX mouse femurs. Here, we confirmed that miR-705 was also overexpressed in osteoporotic mouse mandible BMMSCs. The $q$RT-PCR results showed that miR-705 was significantly upregulated in 3rd-passage OVX mouse mandibular MSCs compared to the expression level in the Sham MSCs (Fig. 3A).

Subsequently, to evaluate whether miR-705 affected osteogenic differentiation in mandibular MSCs, we examined miR-705 expression during osteogenic induction *in vitro*. According to the $q$RT-PCR results, miR-705 expression declined during osteogenic induction, with minimum expression witnessed on day 7 (Fig. 3B). Conversely, its expression increased during adipogenic induction and hit the maximum level on day 7 (Fig. 3C).

## Inhibition of MSC osteoblast lineage commitment by miR-705

To investigate its role in cell lineage commitment, we knocked down and overexpressed miR-705 in mandibular MSCs with the si-miR-705 (inhibitor) and mimics respectively. The effects of the miRNA inhibitor and mimics were confirmed by $q$RT-PCR up to day 14 of induction (Fig. S1A). Alizarin red staining after 14 days of induction confirmed enhanced

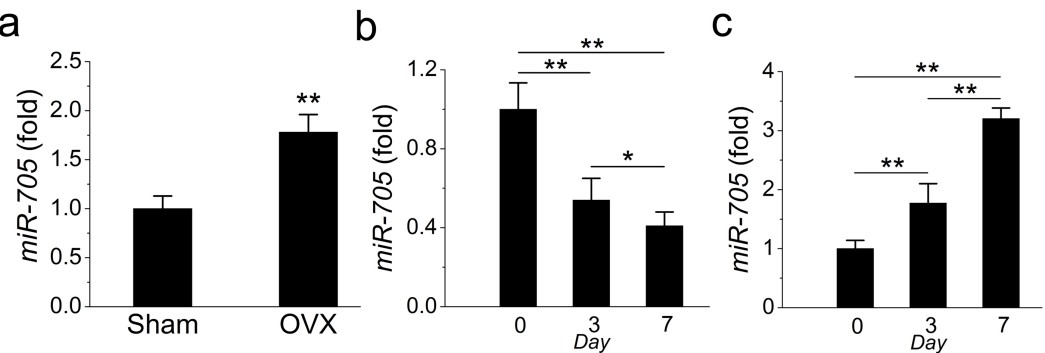

**Figure 3** **miR-705 is enhanced in MSCs from osteoporosis bone marrow.** *q*RT-PCR was used to measure the levels of miR-705 in the third passage of MSCs (A) and at different time point of osteogenic induction (B) and adipogenic induction (C). Values are described as mean ± SD from three independent experiments. *$P < 0.01$, **$P < 0.001$.

nodule formation by mandibular MSCs after miR-705 knockdown (Figs. 4A–4D). And *q*RT-PCR showed enhanced expression of both ALP and RUNX2 in the knockdown MSCs (Figs. 4E, 4F). However, overexpression of miR-705 by the mimics significantly decreased mineralization of the mandibular MSCs as well as the ALP and RUNX2 expression levels.

## Promotion of MSC adipocyte lineage commitment by miR-705

To assess the function of miR-705 in adipocyte lineage commitment, we conducted a gain or loss of function assay on the mandibular MSCs. The effects of the miRNA inhibitor and mimics were confirmed by *q*RT-PCR up to day 7 of induction (Fig. S1B). Oil red O staining showed fewer fat droplets after 7 days of adipo-induction in the miR-705 knockdown MSCs (Figs. 5A–5D). Expression of the adipogenic marker genes LPL and PPAR-$\gamma$ was also decreased on day 7 in the *q*RT-PCR assay (Figs. 5E, 5F). Conversely, promotion of both adipocyte formation and LPL and PPAR-$\gamma$ expression was observed in the miR-705-overexpressing MSCs.

## Knockdown of miR-705 rescues the cell lineage commitment disorder of MSCs from osteoporotic mandibles

Several miRNAs were shown to have functions in shifting MSC linage commitment. In this study, we tested whether knocking down miR-705 attenuated the lineage commitment disorder in MSCs from osteoporotic mandibles. Two groups of OVX mandibular MSCs were transfected either with the miR-705 inhibitor or the transfection reagent alone. And a group of Sham mandibular MSCs served as the control. As shown in Fig. 6, knocking down miR-705 partially rescued osteogenic differentiation of the OVX mandibular MSCs after osteo-induction. Significantly more mineralized nodules were observed in the knockdown group than those in the OVX group (Figs. 6A–6D). This result was confirmed by *q*RT-PCR, which showed increased RUNX2 and ALP mRNA expression in the OVX mandibular MSCs after miR-705 knockdown (Figs. 6E, 6F).

Next, we tested whether the knockdown of miR-705 reduced the excessive adipogenic differentiation of the OVX mandibular MSCs in the same way. The knockdown of miR-705

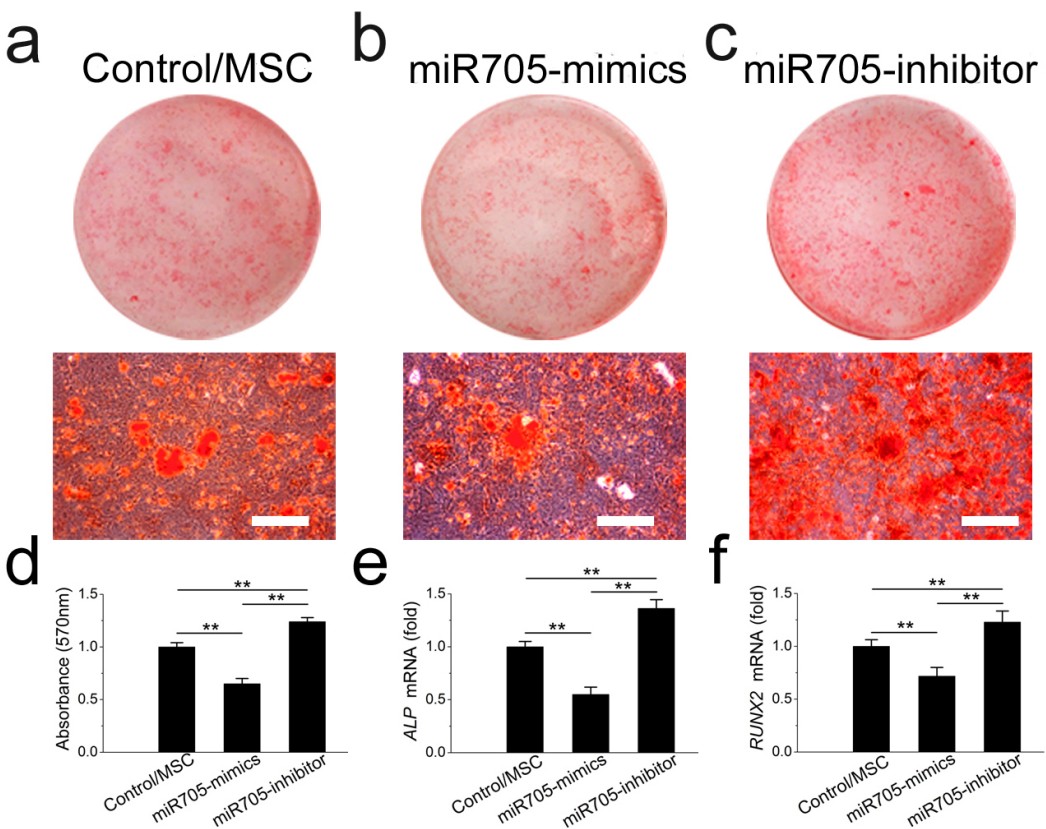

**Figure 4** **miR-705 inhibits osteoblast differentiation of MSCs** MSCs were transfected with miR-705 mimics, inhibitor and the reagent as a negative control. Two days post-transfection, the cells were induced with osteogenic medium for 14 days. After osteogenic induction, alizarin red staining was performed (A–C) and quantified (D). ALP (E) and RUNX2 (F) expression in three groups was measured by $q$ RT-PCR. The scale bars in the micrographs represent 100 μm. Values are described as the mean ± SD from three independent experiments. *$P < 0.01$, **$P < 0.001$.

resulted in reduced lipid droplet accumulation in the induced OVX mandibular MSCs to a certain extent (Figs. 7A–7D). Consistently, it reduced the expression of the adipogenic marker PPAR-$\gamma$ and LPL (Figs. 7E, 7F). These results suggested that the knockdown of miR-705 in OVX mandibular MSCs partially attenuated their adipogenic differentiation.

## Knockdown of miR-705 increases the expression of target genes in MSCs from osteoporotic mandibles

To uncover details of the attenuation observed above, we tested the transcription levels of its predicted target mRNAs HOXA10 and FoxO1 at 48 h post-transfection. As shown in the $q$RT-PCR results, miR-705 expression was remarkably reduced at 48 h after inhibitor transfection (Fig. 8A). According to western blot results, HOXA10 and FoxO1 were suppressed in the MSCs from the OVX mandibles compared with the negative controls but were increased significantly 2 days after miR-705 inhibitor transfection (Fig. 8B).

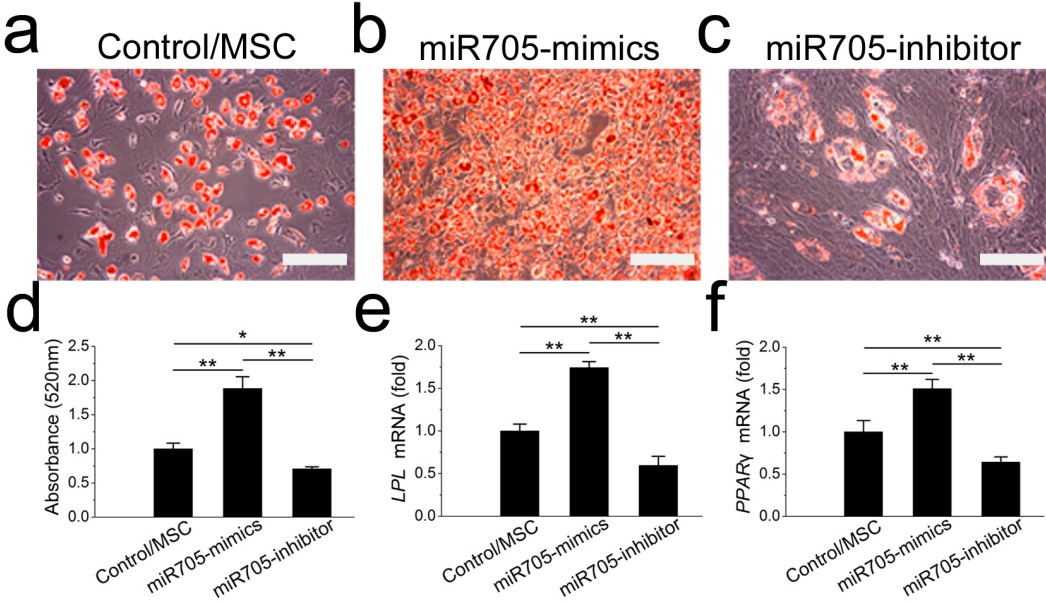

**Figure 5  miR-705 promotes adipocyte differentiation of MSCs.** MSCs were transfected with miR-705 mimics, inhibitor and the reagent as a negative control. Two days post-transfection, the cells were induced with adipogenic medium for 7 days. After adipogenic induction, oil red O staining was performed (A–C) and quantified (D). LPL (E) and PPAR-$\gamma$ (F) expression in three groups was measured by $q$RT-PCR. The scale bars in the micrographs represent 100 $\mu$m. Values are described as the mean $\pm$ SD from three independent experiments. *$P < 0.01$, **$P < 0.001$.

## DISCUSSION

In this study, we cultured MSCs from osteoporotic mouse mandibles and demonstrated that miR-705 acted negatively in mandibular MSC osteoblast but positively in adipocyte lineage commitment. Additionally, inhibiting miR-705 in osteoporotic mandibular MSCs reduced adipogenic differentiation and attenuated the cell lineage commitment disorder.

Studies have reported that systemic diseases, such as osteoporosis and Paget's disease, affect all bones, including alveolar bones (*Galson & Roodman, 2014*). Although the oestrogen deficiency caused by ovariectomy leads to bone mass loss in both craniofacial and long bones, the mandible loses significantly less bone mass than the proximal tibia according to reports from SD rats (*Mavropoulos, Rizzoli & Ammann, 2007*). However, some diseases, such as cherubism (*Brix, Peters & Lebeau, 2009*) and bisphosphonate-related osteonecrosis of the jaws (*Dimitrakopoulos, Magopoulos & Karakasis, 2006*; *Migliorati et al., 2005*), are only witnessed in the craniofacial skeleton. Their distinct responses to external stimuli imply that mandibles may have a different homeostasis mechanism than long bones. Since a disturbance of the alveolar bone marrow may cause many problems to stomatological treatment such as prosthodontics and maxillofacial surgery (*Chen et al., 2016*), obtaining a better understanding of homeostasis in mandibular MSCs will help improve treatment of these diseases.

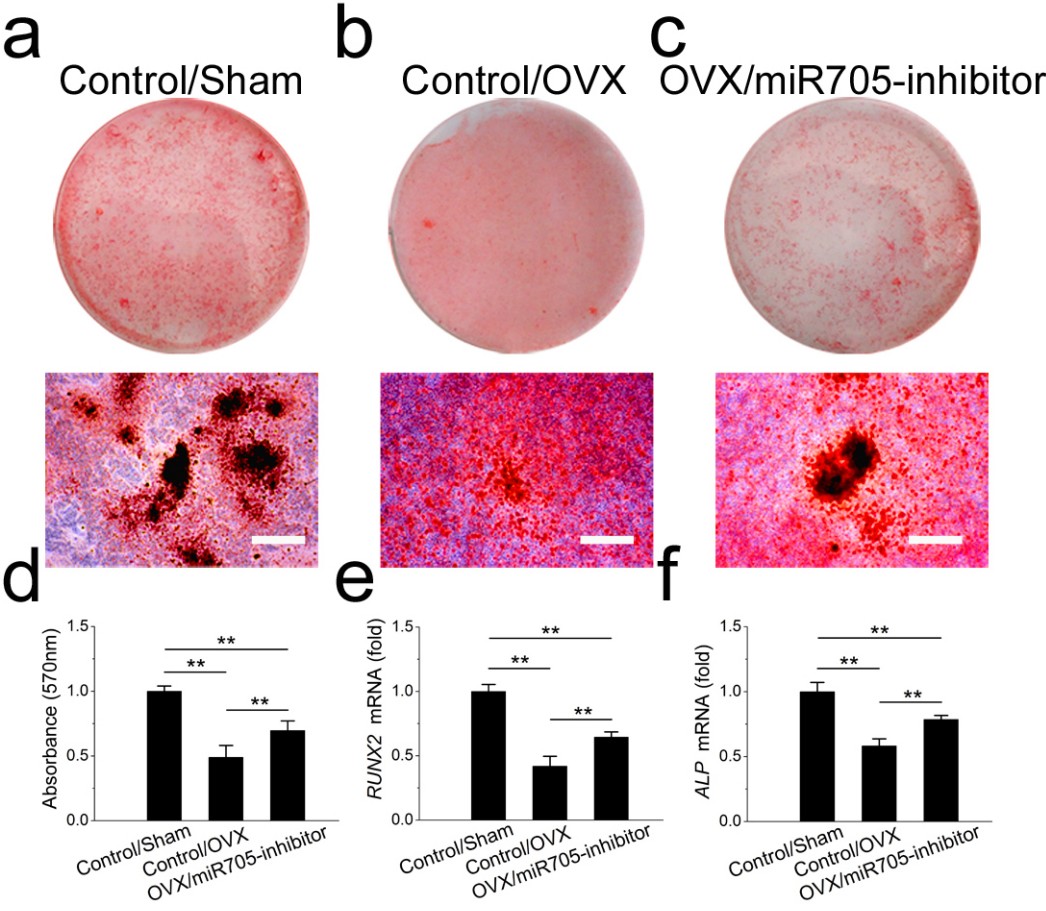

**Figure 6** **Knockdown of miR-705 promotes osteogenic differentiation of OVX MSCs.** Sham MSCs were transfected with the reagent as a negative control. OVX MSCs were transfected with the reagent and miR-705 inhibitor. Three group MSCs were cultured with osteogenic medium for 14 days. After osteogenic induction, alizarin red staining was performed (A–C) and quantified (D). RUNX2 (E) and ALP (F) expression in three groups was measured by $q$RT-PCR. The scale bars in the micrographs represent 100 $\mu$m. Values are described as the mean $\pm$ SD from three independent experiments. $^{*}P < 0.01$, $^{**}P < 0.001$.

In fact, the craniofacial, axial and appendicular skeletons have different developmental origins. The craniofacial structures are primarily formed by neural crest cells, which migrate to the branchial arches during the development process and undergo intra-membranous ossification (*Li, Parada & Chai, 2017*; *Minarikova et al., 2015*; *Minoux & Rijli, 2010*). In contrast, the axial and appendicular skeleton originates from mesenchymal condensations of the mesoderm and adopts endochondral ossification during development (*Kawata et al., 2017*; *Olivares-Navarrete et al., 2017*). All of this evidence suggests that the MSCs in the mandible may play different roles than their counterparts in the long bone marrow.

In this study, we established an osteoporosis model in mice and isolated MSCs from the mandibular bone marrow. To rule out interference of dental pulp and odontoblasts, the periodontal tissue with the whole teeth was removed. Unlike human molars, mouse molars keep erupting for their entire lifespan, thus the force generated during use of

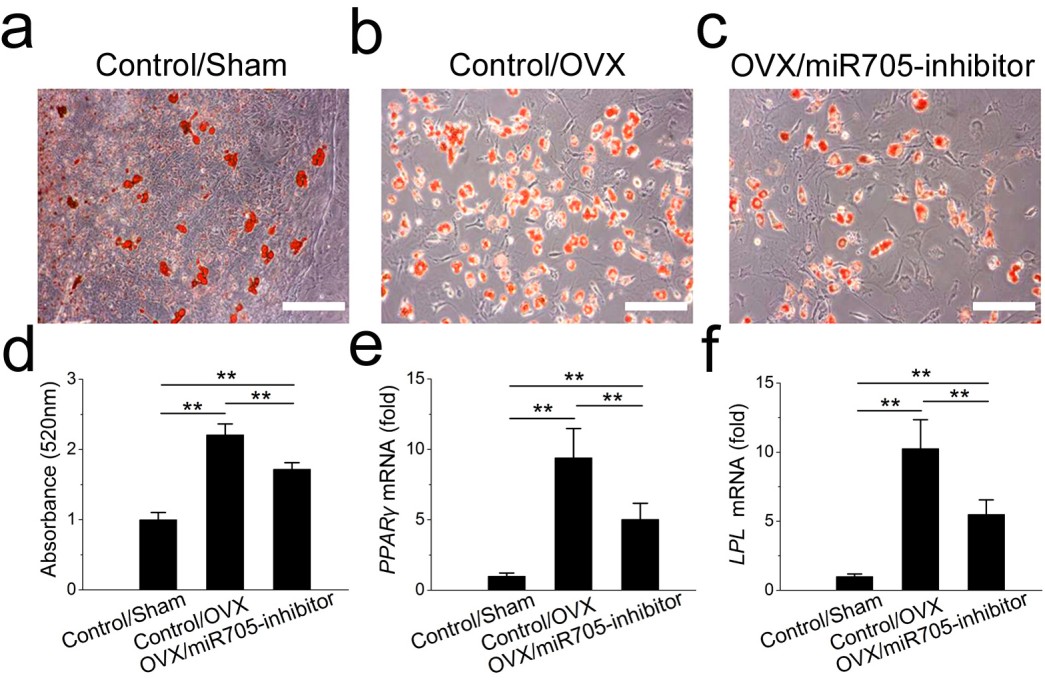

**Figure 7 Knockdown of miR-705 inhibits adipogenic differentiation of OVX MSCs.** Sham MSCs were transfected with the reagent as a negative control. OVX MSCs were transfected with the reagent and miR-705 inhibitor. Three group MSCs were cultured with adipogenic medium for 7 days. After adipogenic induction, oil red O staining was performed (A–C) and quantified (D). PPAR-$\gamma$ (E) and LPL (F) expression in three groups was measured by $q$RT-PCR. The scale bars in the micrographs represent 50 $\mu$m. Values are described as the mean $\pm$ SD from three independent experiments. $^{*}P < 0.01$, $^{**}P < 0.001$.

the alveolar bone beneath the molars keeps changing. To avoid this interference, we chose the rectangular area posterior to the molar as the scan target for micro-CT, which was equivalent to the the angulus mandibulae in the human mandible. As expected, the BMD and BV/TV comparisons were barely significant between the OVX and Sham mouse mandibles, which was quite different from the results obtained from the femur comparisons. The distinct mechanical loading on the mandible may play a role in this protective effect (*Mavropoulos et al., 2014*; *Mavropoulos et al., 2010*; *Mavropoulos, Rizzoli & Ammann, 2007*). Previous studies have shown that mechanical loading on long bones is quite different from that on the mandible and that the mandible is less sensitive to protein under-nutrition or OVX than the proximal tibia spongiosa (*Mavropoulos et al., 2014*; *Mavropoulos et al., 2010*). However, these phenomena were not seen in the MSCs isolated from the OVX mandibular bone marrow which, like the MSCs in the OVX long bones, exhibited weaker osteogenic but stronger adipogenic potential *in vitro*. Analysis of marker gene expression gave similar results. RUNX2 and ALP were expressed at lower levels, but PPAR-$\gamma$ and LPL were expressed at higher levels in the OVX mandibular MSCs than those in the Sham MSCs. The key to the less-sensitive character of the mandible *in vivo* might lay in the unique microenvironment of mandibular MSCs. Therefore, we started with the known clue to identify the main factor in the microenvironment. We focused on miRNAs

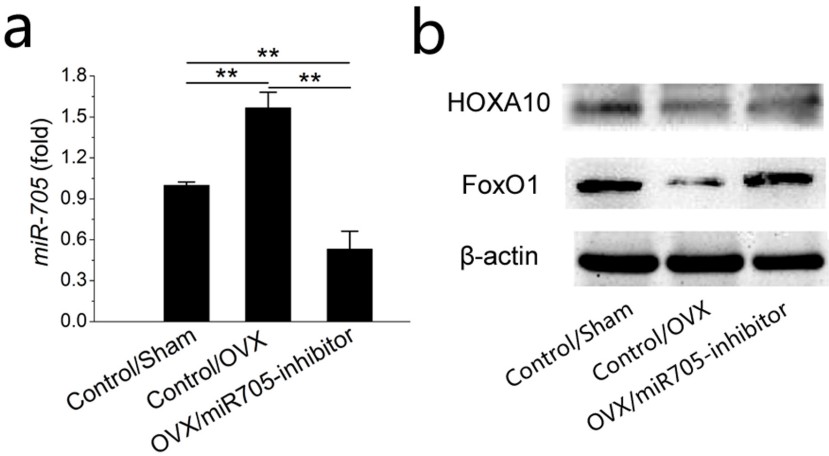

**Figure 8  Knockdown of miR-705 increases the target genes expression in MSCs from osteoporotic mandibles.** Sham MSCs were transfected with the reagent as a negative control. OVX MSCs were transfected with the reagent and miR-705 inhibitor. miR-705 expression in three groups was measured by $q$ RT-PCR at 48 h post-transfection (A). HOXA10 and FoxO1 expression was measured by western blot (B). Values are described as the mean $\pm$ SD from three independent experiments. $^*P < 0.01$, $^{**}P < 0.001$.

that were differently expressed in OVX and Sham MSCs in an attempt to unveil the detailed differences. miRNAs regulate MSC differentiation through the downregulation of target gene expression by either mRNA degradation or translational inhibition (*Ab Mutalib et al., 2016*; *Bartel, 2009*; *Yang, Sui & Liang, 2017*). Their ability to regulate BMMSC lineage commitment has drawn much attention. miR-21, miR-22, miR-27a, and miR-183 have all been reported to regulate the differentiation of BMMSCs (*Gong et al., 2016*; *Ke et al., 2015*; *Yan et al., 2017*; *Zheng et al., 2017*). In a previous study, we demonstrated a regulatory role of miR-705 and miR-3077 in the aetiology of postmenopausal osteoporosis in long bones (*Liao et al., 2013*). Here, we focused on miR-705 to determine whether it regulated mandibular bone marrow MSCs. We found that miR-705 was overexpressed in the MSCs isolated from osteoporotic mouse mandibles. Upregulating miR-705 in mandibular MSCs with mimics resulted in less abundant mineral nodules during osteogenic induction but more lipid droplets during adipogenic induction. Downregulating it with an inhibitor gave opposite outcomes. $q$RT-PCR analysis of both osteogenic and adipogenic marker genes confirmed these results. Thus, we hypothesized that miR-705 played a crucial role in mandibular MSC lineage commitment and mandibular bone marrow homeostasis.

Next, we used the OVX mandibular MSCs to further investigate the role of miR-705 in regulating the homeostasis of these cells. Transfection of the miR-705 inhibitor into OVX mandibular MSCs partially attenuated the impaired osteogenic potential and suppressed the excessive adipogenic differentiation to a certain extent.

These results confirmed our hypothesis that miR-705 inhibited osteoblast differentiation and promoted adipocyte differentiation of mandibular MSCs.

To investigate the mechanism described above, we tested the mRNA expression levels of HOXA10 and FoxO1, which are two important osteogenic genes and predicted targets

of miR-705. Previously, we demonstrated that overexpression of miR-705 suppressed HOXA10 and FoxO1 expression and that they could be partially restored by miR-705 inhibitor transfection (*Liao et al., 2016*; *Liao et al., 2013*). Here we found that miR-705 exerted similarly in OVX mandibular MSCs and that its inhibitor increased HOXA10 and FoxO1 protein expression to a certain extent. However, with numerous predicted target mRNAs, we cannot conclude that miR-705 has the same effects in the mandible and long bones. The mechanism underlying this finding requires more work in the future.

Taken together, our results showed that miR-705 was overexpressed in osteoporotic mandibular MSCs and functioned as a negative regulator of osteoblast differentiation. Suppression of the elevated miRNA by synthetic oligonucleotides may partially rescue the lineage commitment disorder of OVX mandibular MSCs. Utilizing an OVX mouse model, we provided a better understanding of the mechanisms underlying mandibular MSC homeostasis and thus identified a potential therapeutic target for the treatment of mandibular osteoporosis.

## CONCLUSIONS

In conclusion, our study demonstrated that miR-705 was overexpressed in osteoporotic mouse mandibular marrow MSCs and acted negatively in mandibular MSCs osteogenic lineage commitment.

### Funding
This work was supported by grants from the Nature Science Foundation of China (81660180) and the foundation of Hong Hua Gang scientific district (2014–07) and Science and Technology Department of Guizhou Province Foundation (2013–2312). The funders had no role in study design, data collection and analysis, decision to publish, or preparation of the manuscript.

### Grant Disclosures
The following grant information was disclosed by the authors:
Nature Science Foundation of China: 81660180.
Foundation of Hong Hua Gang scientific district: 2014–07.
Science and Technology Department of Guizhou Province Foundation: 2013–2312.

### Competing Interests
The authors declare there are no competing interests.

### Author Contributions
- Xiao Hong Yang conceived and designed the experiments, performed the experiments, contributed reagents/materials/analysis tools, prepared figures and/or tables, authored or reviewed drafts of the paper, approved the final draft.

- Kun Yang and Xiao Hua Hu conceived and designed the experiments, contributed reagents/materials/analysis tools, prepared figures and/or tables, authored or reviewed drafts of the paper, approved the final draft.
- Yu Lin An performed the experiments, prepared figures and/or tables, authored or reviewed drafts of the paper, approved the final draft.
- Li Bo Wang and Guo Luo analyzed the data, authored or reviewed drafts of the paper, approved the final draft.
- conceived and designed the experiments, contributed reagents/materials/analysis tools, prepared figures and/or tables, authored or reviewed drafts of the paper, approved the final draft.

## Animal Ethics

The following information was supplied relating to ethical approvals (i.e., approving body and any reference numbers):

All animal procedures were performed according to the guidelines of the Animal Care Committee of the Zunyi Medical University, Zunyi, Guizhou, China.

## Data Availability

The raw data are provided in the Supplemental Files.

## Supplemental Information

Supplemental information for this article can be found online at http://dx.doi.org/10.7717/peerj.6279#supplemental-information.

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
