# Peer review of "MicroRNA-705 regulates the differentiation of mouse mandible bone marrow mesenchymal stem cells"

_PeerJ, doi:10.7717/peerj.6279_

## Round 0.1 · original submission · Major Revisions

I have attached below the comments from the reviewers for your use in understanding this decision.

You will see that there are some experimental design issues relating to the hypotheses. In addition, both reviewers pointed out that the English language in this submission does not meet our standard. Please be sure to use clear and unambiguous text. All expressions should be grammatically correct and conform to professional standards of courtesy. It is highly recommended to use an English editorial service before submission.

The method for statistical analysis is not appropriate for the data represented in this paper, especially for most of the data consisting three groups. One-way ANOVA followed by multi-comparison test should be performed if the authors want to compare between groups. T-test should be utilized for a statistical comparison of two groups although t-test and ANOVA give the same results.

Reviewer 1 ·

Basic reporting

Introduction
According to Liao et al. 2013, besides miR-705, miR-3077-5p was shown to function as a negative regulatory factor of osteogenic differentiation of BMMSCs derived from long bones of hind limbs. Therefore, the authors should show why they focused on miR-705 not miR-3077-5p in introduction part.

Results
1. In the paragraph from line 185, there were no explanation about Fig. 3c that shows the adipogenic differentiation of mandibular BMMSCs. They should be added in the text. Also, in Fig. 3b and 3c, the results of statistical analysis should be shown.
2. Although the sentences in the line 199-200 and 210-212 showed that the authors analyzed the effect of miR-705 mimic, the results of mimic transfection were not shown in Supplementary Fig. 1a and 1b. They should be added.

Representations and grammars
1. Including the title, there are a lot of phrases “mandible bone marrow” in the text. They should be altered to “mandibular bone marrow”.
2. Throughout the text, en spaces should be added ahead of the brackets to show abbreviations or references.
3. The abbreviations such as “BMMSCs”, “miRNA”, “BMD” and “BV/TV” were restored to their full-spelling in the later part. Once the abbreviations were shown, they should be continuingly used. On the other hand, I think the line 58 “BRONJ” need not to be shown because it was not used subsequently.
4. The phrases “in vitro” and “ex vivo” should be represented in italic letter.
5. Line 35 and 68 “mark genes” would be “marker genes”.
6. Line 39 “osteogenic” would be “osteoporotic”.
7. Line 49 “Despise” would be “Despite”.
8. Line 60 “~in alveolar bones~” would be “~alveolar bones~”.
9. Line 79 “regulation” would be “regulatory”.
10. Line 91 “~anesthesia._After~” would be “~anesthesia. After~”.
11. Line 107 “0.25% trypsin/1m MEDTA” would be “0.25% trypsin/ 1mM EDTA”.
12. Please start a new line ahead of line 123 “Real-time RT-PCR of mRNA and microRNA.”.
13. Please change the font color in the line 129 - 131.
14. Line 134 “Supplementary Table 1” should not be in italic letter.
15. Line 160 “adipocyteduring” would be “adipocyte during”.
16. Line 182 and 201 “---” should be changed to commas.
17. Line 183 “lipoprteinlipase” would be “lipoprotein lipase”.
18. Line 198 “Si-miR-705” would be “si-miR-705”.
19. Line 201 “APL” would be “ALP”.
20. The beginning of the line 183 “705by ~” would be “705 by ~”.
21. Line 210 “gain or loss function” would be “gain or loss of function”.
22. “mir705” and “mir-705” in the line 222, 224, 231, 233 and 234 should be changed to “miR-705”.
23. Line 289 “MiR-21” should be altered to “miR-21”.
24. Line 303 and 304 “Transfection of miR-705 into ~” would be “Transfection of miR-705 inhibitor into ~”.

Experimental design

The results shown were interesting and clear. However, in some part, they didn’t match the hypothesis shown in introduction part (line 62-63). The hypothesis was that some kind of significant roles of mandibular BMMSCs cause the different responses to bone loss stimuli between mandibular and long bones. However, the results shown were not enough to demonstrate what made the differences. To clarify the cause of difference, the responses of mandibular BMMSCs to miR-3077-5p should be examined following the study by Liao et al. 2013. The target mRNA of miR-705 in mandibular BMMSCs should also be analyzed. Such kind of analyses enable the authors to compare with the results in long bones and provide profound information to find the causes of difference.
Considering clinical applications, it is unlikely that miR-705 mimic that might degenerate bone loss would be applied to osteoporosis. However, for comprehensive analysis of the role of miR-705 in bone loss conditions, the effects of miR-705 mimic on the osteogenesis and agipogenesis of mandibular BMMSCs in OVX mouse should be examined.

Validity of the findings

As shown in discussion part (line 276-281), there were different mandibular responses to OVX between in vivo and in vitro. However, I thought that the authors didn't show the consideration of the causes of this difference. Please consider the causes of the difference and add the authors’ interpretation in discussion part.

Additional comments

The authors showed the significant role of miR-705 in lineage commitment of mandibular BMMSCs. I thought that the results shown would bring a lot of important information for the clinical strategies of osteoporosis in the future. However, there were some points to be corrected or added in the manuscript.

·

Basic reporting

The English grammer and proper scientific term (miR/Mir) need to be edited.
The authors owned three out of four manuscript on miR-705 in Pubmed. So,literature and references on miR-705 are very poor.
The premise or significance of this study is not clear and very week
Figures are preliminary, subjective, not mechanistic and very similar kind of experiments.
The quantitative expression of miR-705 is required.How much do they really expressed and how they regulate the differentiation.

Experimental design

Not novel or unique

Validity of the findings

Iam not sure.
Satining and real time PCR are routine analysis.However they are not confirmatory. Western analysis needed.

Additional comments

Fundamentally, the studies mentioned in this manuscript are lacking of miRNA mechanism that regulate the whole process of differentiation.MicroRNAs are post transcriptional repressor through their targets. The authors did not show any of such experiments to prove.Therefore, the studies are very priliminary. More meaningful experiments are needed to show the direct role of miR-705 in the reugulation mandible mesenchymal cell differentiation.

---

## Round 0.2 · Major Revisions

I have attached below the comments from the reviewer for your use in understanding this decision.

You will see that this manuscript still suffers from several issues including English grammar, punctuation and, most importantly, the hypothesis of this study.

Sufficient information about the method for statistical analysis is still not provided. There are many kinds of multi-comparison test utilized for a statistical comparison of the three groups. Please specify the method utilized for multi comparison.

Reviewer 1 ·

Basic reporting

Representations and grammars
 As I pointed out in the first review, en spaces should be added ahead of the brackets to show abbreviations or references. Also, just after a bracket and a period such as line 79 and 116 respectively, en spaces should be inserted. Thus, lots of mistakes of the use of punctuation are found throughout the text. I highly recommend the authors to get English proofreading.
 Line 21 and 64 “(maxillary)” would be “or maxillary” or “and maxillary”.
 The underlines drawn in the line 264 and 265 should be deleted.
 Line 281 “haemostasis” would be “homeostasis”.
 The phrases “in vivo” in the line 316 should be represented in italic letter.

Results
I found the authors have added the explanation of Fig. 3c in the revised text. However, the results of statistical analysis are not still shown in revised Fig. 3b and 3c. The expression levels of miR-705 in each time point seem to be significantly different. If they are not significantly different, they should be shown as such in figures and their legends.

Experimental design

No comment.

Validity of the findings

What is the hypothesis of this study? The hypothesis shown in the line 63-65 and those in the line 334-336 and 342-343 are similar, however, they are not the same. According to the hypothesis shown in the line 63-65, the authors think that some kind of significant roles of mandibular BMMSCs cause the different responses to bone loss stimuli between mandibular and long bones. On the other hand, the experiments conducted in this study are for the analysis of the roles of miR-705 in mandibular BMMSCs, not to analyze the roles of mandibular BMMSCs to cause the different responses between mandibular and long bones. The analysis performed in this study rather matches the hypothesis shown in the line 334-336 and 342-343. Therefore, although it is undesirable, the hypothesis shown in introduction part should be arranged to match conducted analyses.

Additional comments

I thank the authors for correcting their manuscript. Especially, in the discussion part, the authors added lots of information from other related studies, and the authors showed their considerations for additional experiment and our suggestions. Also, I understood the authors’ situations concerning the analysis of miR-3077-5p. I look forward to see the findings in the future. However, I found some points to be corrected to improve the authors’ study.

---

## Round 0.3 · Major Revisions

1. The statistical issue.

Basically, Fisher's original post-hoc test, the LSD, has the problem that it produces inaccurate p-values by performing multiple t-tests on the same sample. This leads to a situation where the false-discovery rate increases with each pair of means that is compared. 

With three groups, you have three possible comparisons, giving you a false-discovery rate of 14% (recall that the false-discovery rate should be 5%).

So the LSD is fatally flawed, and should not be used at all. 

You can read about the LSD here.
https://www.graphpad.com/guides/prism/7/statistics/index.htm?stat_fishers_lsd.htm

I personally recommend the Tukey-Kramer test as a post-hoc.
Please revise the manuscript by utilizing this method.
Please note that you may see some differences in the statistical results, i.e. some comparisons between groups may become statistically not significant.
You may need to change the method, results, figure, and/or discussion according to the new test.

In addition, most importantly, please upload the raw data for statistical calculation showing the results from LSD and Tukey-Kramer test, so that I can understand what kind of differences could be or could not be found between LSD and Tukey-Kramer test.

2. The English issue.
As the reviewer pointed out several times, the English in this manuscript definitely needs to be improved.

As a condition of continued consideration, I inform you that if the required *substantial* improvements do not happen in the next revision, I would be forced to reject your manuscript.

Reviewer 1 ·

Basic reporting

As I pointed out many times over, there are still lots of adjusting points related to punctuations and use of abbreviations. For instance, in the introduction and discussion parts, absence of en spaces are recognized a lot ahead of the brackets. In the line 93-94, “OVX” is shown as a new abbreviation though it has already represented above in the same sentence.
I assume that there are similar adjusting points concerning representations more. I recommend the authors to get English proofreading again.

Experimental design

No comment.

Validity of the findings

No comment.

Additional comments

No comment.

---

## Round 0.4 · accepted · Accept

Your manuscript has been improved tremendously and deserves to be accepted.

Reviewer 1 ·

Basic reporting

No comment.

Experimental design

No comment.

Validity of the findings

No comment

Additional comments

The adjusting points that we pointed out previously were corrected properly in the revised version. As a result, I think the overall quality of the authors’ work has been remarkably improved. I would commit the final decision to the editor.